# Expression of Maize MADS Transcription Factor *ZmES22* Negatively Modulates Starch Accumulation in Rice Endosperm

**DOI:** 10.3390/ijms20030483

**Published:** 2019-01-23

**Authors:** Kangyong Zha, Haoxun Xie, Min Ge, Zimeng Wang, Yu Wang, Weina Si, Longjiang Gu

**Affiliations:** National Engineering Laboratory of Crop Stress Resistance breeding, Anhui Agricultural University, Hefei 230036, China; kangyongzha929@163.com (K.Z.); xhx521xz@163.com (H.X.); 13215616815@163.com (M.G.); wzmhanchang@163.com (Z.W.); wangyu20180712@163.com (Y.W.); weinasi@ahau.edu.cn (W.S.)

**Keywords:** *Zea mays* L., MADS transcription factor, *ZmES22*, starch

## Abstract

As major component in cereals grains, starch has been one of the most important carbohydrate consumed by a majority of world’s population. However, the molecular mechanism for regulation of biosynthesis of starch remains elusive. In the present study, *ZmES22*, encoding a MADS-type transcription factor, was modestly characterized from maize inbred line B73. *ZmES22* exhibited high expression level in endosperm at 10 days after pollination (DAP) and peaked in endosperm at 20 DAP, indicating that *ZmES22* was preferentially expressed in maize endosperm during active starch synthesis. Transient expression of *ZmES22* in tobacco leaf revealed that ZmES22 protein located in nucleus. No transactivation activity could be detected for ZmES22 protein via yeast one-hybrid assay. Transformation of overexpressing plasmid 35S::*ZmES22* into rice remarkedly reduced 1000-grain weight as well as the total starch content, while the soluble sugar was significantly higher in transgenic rice lines. Moreover, overexpressing *ZmES22* reduced fractions of long branched starch. Scanning electron microscopy images of transverse sections of rice grains revealed that altered expression of *ZmES22* also changed the morphology of starch granule from densely packed, polyhedral starch granules into loosely packed, spherical granules with larger spaces. Furthermore, RNA-seq results indicated that overexpressing *ZmES22* could significantly influence mRNA expression levels of numerous key regulatory genes in starch synthesis pathway. Y1H assay illustrated that ZmES22 protein could bind to the promoter region of *OsGIF1* and downregulate its mRNA expression during rice grain filling stages. These findings suggest that *ZmES22* was a novel regulator during starch synthesis process in rice endosperm.

## 1. Introduction

Maize (*Zea mays* L.) is one of the most widely grown crop world-wide, as well as a critical model for various biological researches, especially for endosperm development [1]. Starch is the major component of maize grains, which accounted up to 71% on a dry weight basis. Therefore, comprehensive understanding of the molecular mechanism for regulation of starch synthesis will facilitate increase in yield to feed growing population.

Starch is composed of two major components, known as amylose and amylopectin. The process of starch biosynthesis has been reported to be under finely regulated by numerous genes, which mainly encoded multiple subunits or isoforms of four enzymes: ADP-glucose pyrophosphorylase (AGPase), starch synthase (SS), starch branching enzyme (SBE), and starch debranching enzyme (DBE) [2,3]. At the initial stage of starch synthesis, glucose-1-phosphate, together with ATP, are converted to ADP-glucose (ADPG) via AGPase. In the developing endosperm, ADPG is mainly produced in the cytosol and transferred into amyloplast through an adenylate translocator, BT1 [4]. Afterwards, the synthesis of starch is furthered by chain elongation by transferring ADPG to the nonreducing end of a glucan primer. The amylose chain elongation is completed by granule-bound starch synthase I (GBSSI), whereas, amylopectin chains are elongated by a soluble form of starch synthase (SSI, SSII, SSIII, and SSIV). α-1,6-Glucosidic linkages is then introduced by starch branching enzyme (BEI and BEII) and finally, fine structure of amylopectin is achieved through removal of unnecessary branches by starch debranching enzymes (ISA and Pullulanase). Mutants defective in any key genes exhibited apparent abnormal characters of starch in reserve organs. Mutations in *OsAGPL2*, one of the large subunits of AGPase, caused severe defects in grain filling and starch synthesis [5]. Loss-of-function mutations occurred in *OsBT1* gene, which encoded an ADPG translocator, resulted in a remarkable reduction in grain weight than wild type [4]. Deficiency of *OsSSIIa* lead to a chalky interior appearance and the endosperm of the mutant lines are mainly consisted of loosely packed, spherical starch granules with larger air spaces [5]. *Grain Incomplete Filling 1* (*OsGIF1*), encoding a cell-wall invertase, was of great importance in regulation of sucrose unloading from phloem into cells of reserve organs. Mutant lines of *OsGIF1* showed severe defects in grain filling and in turn reduced the grain weight to 70% of wild type rice at 30 days after pollination (DAP) [6].

Since starch biosynthesis and accumulation are critical determinants for both grain quality and production, key transcriptional regulators, including several transcription factors (TFs), have also been demonstrated to play an important regulatory role in starch synthesis. Null mutants of *OsBZIP58* seeds exhibited altered starch composition as well as morphological defects with apparent white belly region [7]. SUSIBA2, a WRKY family transcription factor, could directly bind to the promoter of *pISA1* gene to regulate its expression, thus affecting the synthesis of starch in barley [8]. Additionally, one of AP2 family of transcription factors, SERF1, negatively regulates rice grain filling, and genetic mutations could enhance the starch synthesis process of rice [9]. *ZmbZIP91* was proved to be a key regulator of the starch synthesis by directly binding to ACTCAT elements in the promoters of starch synthesis genes [10]. The inhibition of *ZmDof3* led to defects of the kernel phenotype with decreased starch content and a partially patchy aleurone layer [1]. Altered expression of transcription factors, causing abnormal features in reserve organs, could provide profound implications in understanding the molecular mechanisms that control starch biosynthesis. Despite these research highlights, a comprehensive understanding of factors that regulate the expression of genes in network of starch synthesis remains largely unknown, especially in maize. Hence, screening and identification of key transcription factors involved in starch synthesis will be of great importance in breeding of high-yielding crops.

In previous studies, a total of 2298 transcription factors were identified and further examined using RNA-seq dataset from 18 representative tissues from maize [11], which provided profound clues regarding to the relationship between development and dynamic expression profiles of key transcription factors. With an emphasis on endosperm-specificity, we identified 36 transcription factors that were preferentially highly expressed in maize endosperm [12]. The mRNA expression profiles of one gene, encoding a typical MADS transcription factor (GRMZM2G159397, designated as *ZmES22*), were further confirmed via qRT-PCR assays. To test if this gene was related to starch synthesis, *ZmES22* was cloned from maize inbred line B73. Afterwards, molecular properties and biological functions were modestly comprehensively characterized in transgenic rice lines. Overexpressing *ZmES22* in rice significantly reduced 1000-grain weight as well as hindered starch accumulation. Besides, altered expression of *ZmES22* in transgenic rice also changed the starch structure and morphology of starch granules. Furthermore, RNA-seq analysis demonstrated that numerous key regulatory genes in starch synthesis were differentially expressed compared to that in WT plants. Yeast one hybrid assay revealed that ZmES22 could bind to the promoter of *OsGIF1* and downregulated its expression during grain filling process. This study illustrated that *ZmES22* could be a newfound transcription factor, which negatively regulated starch synthesis in rice endosperm.

## 2. Results

### 2.1. Sequence Analysis and Construction of Phylogenetic Tree for ZmES22 Homologues

As one of the largest transcription factor family in eukaryote, MADS-box proteins has been characterized by its important roles in a variety of aspects during plant growth and development [13,14]. To test if *ZmES22* were related to starch synthesis, this gene was firstly cloned from maize inbred line B73. *ZmES22* contained an open reading frame (ORF) of 723 bp and encoded a protein of 240 amino acids with a predicted molecular weight (Mw) of 27,903 Da and an isoelectric point (pI) of 8.92. Pfam analysis of ZmES22 revealed that the deduced protein sequence consisted of four conserved domains, namely the MADS-box domain (MADS-box), intervening (I), K-box domain, and the C terminal domain (Appendix A). In order to find homologs of *ZmES22*, blastp program was explored for protein sequence of ZmES22 to search against protein database for *Zea mays*, *Oryza sativa*, and *Arabidopsis thaliana*, respectively. Afterwards, pairwise amino acid distances were calculated using MEGA7 with Jones–Taylor–Thornton (JTT) model, and genes with diversity less than 0.8 were retained according to empirical experience. A total of 16 genes, including 6 from *Zea mays*, 5 from *Oryza sativa*, and 5 from *Arabidopsis thaliana* were identified, respectively (Appendix A). Phylogenetic tree was constructed using the conserved MADS domain, and clear orthologous relationship could be observed between *ZmES22* and *OsMADS7* (Figure 1 and Appendix A). Furthermore, the Multiple EM for Motif Elicitation (MEME) motif website search program was explored to identify the conserved motifs for all 16 homologues. Great majority homologues contained Motif 1, Motif 2, Motif 3, and Motif 4, indicating these motifs were probably evolutionary conserved (Figure 1). While, presence or absence for remained motifs was more variable. 

### 2.2. Expression Profiles and Subcellular Localization of ZmES22

qRT-PCR assays were performed to investigate expression profiles of *ZmES22*. In line with previous transcriptome analysis, compared with nutritive organs, such as root, stem and leaf, *ZmES22* exhibited higher relative expression levels in reproductive organs (Figure 2). Intriguingly, significantly higher expression level of *ZmES22* was observed in endosperm than embryo at 10 DAP, and mRNA expression of *ZmES22* peaked in endosperm at 20 DAP. To ascertain the location of *ZmES22* protein, coding sequence of *ZmES22* was inserted into empty vector 35S::GFP. Afterwards, 35S::*ZmES22*-GFP construct and 35S::GFP were efficiently transfected tobacco leaf cells separately via *Agrobacterium* infiltration (Figure 3). Green fluorescence of 35S::GFP could be observed throughout the cell, whereas, green fluorescence of ZmES22-GFP fusion protein appeared only in nucleus (Figure 3), illustrating that ZmES22 protein functioned in nucleus. However, yeast one-hybrid assay demonstrated that ZmES22 protein did not have transcriptional activity in yeast cells (Appendix A). These results indicated that *ZmES22* may be involved in the regulation of development of endosperm with the help of other proteins.

### 2.3. Analysis of Agronomic Characters of ZmES22 Overexpression Transgenic Rice

To illustrate the function of *ZmES22*, twelve independent rice lines, which overexpressed *ZmES22* under the drive of CaMV 35S promoter, were obtained via *Agrobacterium* mediated transformation. qRT-PCR assays revealed that *ZmES22* expressed at distinct levels in transgenic rice lines, among which L8, L9, and L10 exhibited significantly higher expression (Appendix A). Therefore, these three transgenic rice lines was selected for further research. Compared to wild type (WT) plants, overexpression rice lines exhibited no visible difference during both the vegetative and reproductive stages, with similar plant height as well as panicle architecture (Appendix A). After maturation, agronomic traits, including grain length, grain width, grain thickness, and 1000-grain weight were minutely characterized for both transgenic rice lines and WT plants. There was no significant change in either grain length or grain width between transgenic plants and WT plants (Figure 4A,B). Nevertheless, grain thickness was dramatically decreased in overexpressed rice lines (Figure 4C, Student’s *t*-test, *p*-value = 4.8 × 10^−5^). Accordingly, 1000-grain weight of transgenic plants were significantly depleted by 3.88 g than that of WT plants (Figure 4D, Student’s *t*-test, *p*-value = 1.8 × 10^−8^). Additionally, total starch content, apparent amylose content (AAC) and soluble sugar content of both transgenic rice lines and WT plants were measured according to previously reported methods. Surprisingly, compared to WT plants, both total starch content and AAC were significantly reduced (Figure 5A,B, Student’s *t*-test, *p*-value = 0.02), whereas, the content of soluble sugar in transgenic rice lines were significantly increased by 38% than that of WT plants (Figure 5C, Student’s *t*-test, *p*-value = 8.4 × 10^−4^). In particular, content of soluble sugar was two times larger in transgenic line L9 than that in WT plants. These results revealed that overexpression of *ZmES22* gene could significantly block starch biosynthesis process in endosperm of rice.

### 2.4. Overexpression of ZmES22 Influences Starch Structure in Transgenic Rice

Both amylopectin blue value and the maximum absorption wavelength reflect the ability of amylopectin binding to iodine. Therefore, different BV and kmax can provide indicators for the basic distinction of starch structure [15]. To detect whether the relative content of amylose and amylopectin were altered by overexpression of *ZmES22*, BV and kmax of starch from both *ZmES22* overexpression rice seeds and WT plants were determined accordingly. As shown in Figure 6A,B, both BV and kmax of amylose and amylopectin in three transgenic lines were significantly smaller than that of WT plants (Student’s *t*-test, *p*-value = 2.2 × 10^−16^). Furthermore, morphology of starch granules was examined via scanning electron microscopy (SEM) [16]. SEM images of transverse sections of rice grains revealed that both central and dorsal endosperms were filled with densely packed, polyhedral starch granules in both transgenic rice and WT seeds, while ventral endosperm of transgenic rice seeds exhibited an apparent abnormity with a visible chalky region (Figure 6C), which was mainly consisted of loosely packed, spherical starch granules with larger air spaces. These results indicated that the overexpressing *ZmES22* could change starch structure as well as influence morphology of starch granules in transgenic rice lines.

### 2.5. Overexpression of ZmES22 Influence Expression Profiles of Numerous Starch Synthesis Related Genes at 20 DAP Endosperm

To further explore the molecular basis of *ZmES22* in regulation of starch synthesis, expression profiles of 17 genes, which were preferentially expressed in developing endosperms and were demonstrated to be involved in starch synthesis, were compared between transgenic rice lines with WT plants at different developmental stages (3, 6, 10, and 20 DAP, Figure 7). The results illustrated that, except for *OsISA2*, *OsSSI*, and *OsSSIIa*, great majority of characterized starch synthesis related genes were downregulated depending on the individual genes when compared to WT plants (Figure 7). Interestingly, expression levels of *OsBEI* and *OsPUL* exhibited similar tendency that they were remarkably upregulated as grains got maturity (Figure 7). As is described previously, *ZmES22* was highly expressed in 20 DAP endosperm, therefore, we proposed that genes differentially expressed in transgenic rice plants at 20 DAP endosperm might be potentially key regulators. Nevertheless, no significant changes could be observed among all of tested genes at 20 DAP endosperms (Figure 7). In order to further investigate possible regulation by *ZmES22*, 20-DAP seeds for both overexpression rice lines and WT plants were collected for RNA-seq analysis, each was repeated with two biological replicates (Table 1, Figure 8A,B). Collectively, 1902 differentially expressed genes (DEGs), consisting of 986 upregulated and 916 downregulated genes in overexpression rice lines (Figure 8C), were determined with the following criteria:(1) the minimum fold-change of gene expression was 2.0; (2) the maximum adjusted *p* value was 0.05. In order to validate the RNA-seq data, 10 DEGs, including 5 upregulated and 5 downregulated genes, were randomly selected for quantitative real-time PCR analysis, and the results illustrated that RNA-seq data are of satisfactory quality (Figure 8D). To analyze the functional enrichments of the DEGs, both Gene Ontology (GO) and Kyoto Encyclopedia of Genes and Genomes (KEGG) analysis were performed using R package ClusterProfiler. DNA metabolic process, response to stress and carbohydrate metabolic process are the three mostly enriched GO items (Appendix A). Moreover, six pathways were significantly enriched in KEGG analysis (Figure 9), including starch and sucrose metabolism pathway (Appendix A), galactose metabolism, phenylalanine metabolism, plant hormone signal transduction pathway (Appendix A), etc. Interestingly, one gene, named *GIF1* (*Os04g0413500*), which was reported to be a key regulator to rice grain-filling and yield, was significantly enriched in carbohydrate metabolic process as well as starch and sucrose metabolism pathway in KEGG. The *gif1* mutant exhibited slower grain-filling rate and showed markedly more grain chalkiness than wild-type plants [6]. In the present study, *GIF1* gene was downregulated as much as 8-fold in overexpression rice lines compared with WT plants (Fisher’s exact test, *p* = 5.0 × 10^−5^). The relative mRNA expression level of *OsGIF1* in four different developmental endosperms (3, 6, 10, and 20 DAP) were further confirmed by real-time quantitative PCR (Appendix A). Because overexpression *ZmES22* lead to similar phenotype as *gif1* mutant, we therefore wonder if ZmES22 could bind to the promoter of *GIF1* and negatively regulate its expression? 

### 2.6. ZmES22 Could Bind to Promoter GIF1 Gene

To determine if ZmES22 could bind to the promoter of *GIF1*, we firstly extracted 2000 base-pair sequences from upstream of *GIF1* and subjected it web site (http://bioinformatics.psb.ugent.be/webtools/plantcare/html/) to predict if promoter of *GIF1* contained conserved element that MADS type transcription factors could bind to [17]. Surprisingly, a conserved element (CATGT) was located at minus 365 base-pair in upstream of *OsGIF1* gene [18] (Appendix A). Afterwards, yeast one hybrid assay was explored to determine whether ZmES22 protein could bind to this element. Complete coding sequence of *ZmES22* was inserted into vector containing both activation domain (AD) and was drove by P_T7_. Two repeated copies of CATGT was synthesized as bait (designated as pGIF1). Simultaneously, a mutant (CAGGT, designated as pmGIF1) was also used as a negative control. As was illustrated in Figure 10, both the growth of the yeast in null and negative control were obviously inhibited in SD/-Ura medium with 900 ng/mL AbA in the yeast one-hybrid assay, however, the yeast co-transformed with P_T7_-*ZmES22* and pGIF1-AbAi grows well, indicating that ZmES22 binds to the core element of the promoter of *OsGIF1* (Figure 10).

## 3. Discussion

The endosperm is the tissue that most flowering plants produce in the seeds after fertilization. Endosperm development involves the process of starch synthesis and storage protein accumulation. Recent studies revealed that process of starch synthesis was remarkably conserved ranging from green algae to extant higher plants, suggesting that genes encoding starch biosynthesis related enzymes were functionally conserved across diverse lineages [19,20]. To date, enzymes involved in starch synthesis has been soundly documented in rice. Therefore, rice endosperm is a particularly ideal model to screen and identify key transcription factors that could finely tune the process of starch synthesis in maize [21].

In the present study, transgenic rice that overexpressed one MADS type transcription factor *ZmES22* from maize, exhibited obvious defects with respect to grain characteristics, represented by loosely packed starch granules, reduced 1000-grain weight, and altered apparent amylose and total starch content, suggesting that *ZmES22* might play a key role in regulation of starch synthesis pathway in the transgenic lines. It has been reported that the process of starch biosynthesis was regulated by 17 genes, which mainly encoded multiple subunits or isoforms of four enzymes: ADP-glucose pyrophosphorylase (AGPase), starch synthase (SS), starch branching enzyme (SBE), and starch debranching enzyme (DBE) [2,3]. Interestingly, no significant expression change could be detected for all of these 17 genes in endosperm at 20 DAP between wide type and transgenic rice based on clues from both qRT-PCR and RNA-seq results. However, the mRNA expression levels of majority of starch synthesis related genes, except for *OsISA1*, *OsISA2*, *OsSSI*, and *OsSSIIa*, decreased during the early stages of endosperm development stages compared to that in wild type rice. These results indicated that overexpression of *ZmES22* could negatively affect mRNA expression of the majority of starch synthesis related genes in distinct degree during the early endosperm development stages.

KEGG analysis revealed that DEGs was significantly enriched in starch and sucrose metabolism pathway, in which the mRNA expression of one gene, named *OsGIF1*, decreased as much as 8-fold in transgenic rice. Previous result demonstrated that grain filling rate of *gif1* mutants was slower and accompanied with distinct chalkiness and loosely packed starch granules, while overexpression of *GIF1* driven by its native promoter produces larger grains [6]. The phenotype of *gif1* mutant was consistent to transgenic rice lines that overexpressed *ZmES22* gene from maize. Evidence from qRT-PCR also validated that mRNA expression level of *OsGIF1* continually decreased in overexpression rice lines in comparison to wild type plants. These results indicated that overexpression of *ZmES22* in rice might inhibit the mRNA expression of *OsGIF1*. As is reported that typic MADS type transcription factor are plant specific and often contained four functional domains, the MADS-box conserved domain (MADS-box), intervening (I), K-box domain, which is homologous to keratin (K), and the C terminal domain. MADS-box domain could bind to the promoter and regulate the expression of downstream genes. For example, *ZmMADS47* directly binds the core motif CATGT of promoter of zein genes and activated its expression [18]. Additionally, CATGT element also resided in the upstream of transcription starting sites (−365 base pair), yeast one hybrid assay demonstrated that ZmES22 could bind to the core motif of *OsGIF1* and repress its expression. 

The primary results provide evidence that *ZmES22* affect starch synthesis and endosperm development through binding to and downregulating the expression of *OsGIF1*, and in turn influencing carbon distribution and transportation of sucrose on grain filling in rice plant.

In conclusion, a MADS type transcription factor from maize was modestly characterized via overexpression in rice and the molecular mechanism for its anticipant role in regulation of starch synthesis were also explored by RNA-seq and yeast one hybrid assay in the present study. Starch synthesis is a complicated and sophisticated process, which is regulated by numerous transcription factors via protein–protein and protein–DNA interactions. In order to shed light on how *ZmES22* influence the starch synthesis in rice, *ZmES22* mutant are being created via CRISPR/Cas9 system.

## 4. Materials and Methods

### 4.1. Plant Materials and Growth Conditions

Ten representative tissues, including root, stem, leaf, tassel, filament, ear, 10 DAP (days after pollination) embryo, 20 DAP embryo, 10 DAP endosperm, and 20 DAP endosperm, were collected from maize inbred line B73 plants, which were grown in a greenhouse with paddy soil at 28 °C under a 14 h light/10 h dark photoperiod. Each tissue was repeatedly sampled from three individual plants as three biological replicates. Both wild type cultivars (*Oryza sativa* L. *japonica* cv. Zhonghua 11) and transgenic rice lines were grown under natural conditions in experimental field plots for Anhui Agricultural University in Anhui Province, China. Rice endosperms at 3, 6, 10, and 20 DAP were harvested for qRT-PCR assay.

### 4.2. RNA Extraction and Real-Time RT-PCR Analysis

Total RNA of collected samples was extracted using RNAiso Plus Kit (Takara, Kusatsu, Japan) according to manufacturer’s instructions. Afterwards, first-strand cDNA was generated using reverse transcription system (Promega, Madison, WI, USA). The qRT-PCR (quantitative real-time PCR) was performed using SYBR Green Master (Roche, Basel, Switzerland) on an ABI 7300 Real Time PCR System (Applied Biosystems, Foster City, CA, USA), and the reactions were performed according to previous report [22]. In detail, the reaction conditions were set as following: 50 °C for 2 min, then 95 °C for 10 min, followed by 40 cycles of 95 °C for 15 s and 60 °C for 1 min. Each cDNA sample was quantified in three replicates. The achieved data were calculated by 2_DDCt method as described previously [23]. Maize *Actin1* gene was used as internal control to normalize the detection threshold for each of three replicates.

### 4.3. Subcellular Localization

Full-length open reading frame (ORF) of *ZmES22* without the termination codon was amplified and cloned into pCAMBIA1305 vector under the drive of cauliflower mosaic virus (CaMV) 35S promoter. The fusion construct 35S:: *ZmES22*-GFP and empty vector 35S:: GFP were transformed into leaves for 35-day-old tobacco (*Nicotiana bethamiana*) via a syringe without needle, respectively. Infiltrated tobacco plants were transferred into dark condition for 12 h, followed by normal illumination for 48 h. Finally, green fluorescent signals were examined using confocal microscope (Olympus FV1000, Tokyo, Japan).

### 4.4. Transcriptional Activation Assay

Full-length ORF of *ZmES22* was amplified and inserted into pGBKT7 vector (Clontech, San Deigo, CA, USA), which were fused with the GAL4 DNA-binding domain beforehand. Subsequently, both negative control (null pGBKT7 vector) and positive control (co-transformation of pGBKT7-53 with pGADT7-T vectors), together with fusion construct pGBKT7-*ZmES22* was transformed into yeast strain AH109, respectively. AH109 strain carries HIS3, ADE2 and MEL1 reporter genes. Transformed yeast cells were then cultured on SD/-Trp medium for 3 days at 30 °C and then transferred into SD/-Trp/-His/-Ade/X-α-GAL medium for 3 days at 30 °C.

### 4.5. Generation of Transgenic Rice Lines

Full-length ORF of *ZmES22* was amplified and inserted into an overexpression vector pCAMBIA1301a under the drive of CaMV 35S promoter and a NOS terminator. Recombination construct pCAMBIA1301a-*ZmES22* also harbored a GUS reporter gene and was transformed into *japonica* rice cultivar Zhonghua 11 via *Agrobacterium* mediated transformation [24]. Both histochemical staining of GUS activity and PCR experiments followed by sanger sequencing were utilized to validate if transgenic rice lines were positive ones.

### 4.6. Determination of Agronomic Characters and Measurement of Grain Quality

Vernier caliper was adopted to measure the length, width and thickness for 100 uniformly mature seeds at the longest, widest, and thickest point, respectively. 1000-grain weight was determined by counting ten independent repeats of 100-grain samples on an electronic balance. Each measurement was repeated for three times. Embryos and follicles were separated from the embryo and ground into powder. The starch content was measured with total starch determination kit (K-TSTA; Megazyme, Bray, County Wicklow, Ireland) according to manufacturer’s protocol. Apparent amylose content (AAC) of the samples was measured using iodine colorimetry (K-AMYL; Megazyme) [15]. Anthrone method was applied to determine soluble sugar content [7].

### 4.7. Measurement of Starch Blue Value (BV) and Maximum Absorption Wavelength (kmax)

The separation of amylose and amylopectin, and measurement of the blue value (BV) and maximum absorption wavelength (kmax) of starch were referred to the modified alkali impregnation method [25]. Detailedly, 5 mg isolated amylose powder was dissolved in 8 mL 90% dimethyl sulfoxide and then diluted to 100 mL double distilled water. The absorption spectra of the starch-iodine complex were examined ranging from 500 to 800 nm. The BV was A_600_. While, with respect to amylopectin, 15 mg isolated amylopectin powder was dissolved in 100 mL double distilled water. The absorption spectra were examined ranging from 500 to 700 nm, and the BV was set to A_680_.

### 4.8. Observation of Starch Granules by Scanning Electron Microscopy (SEM)

According to the methods in previous report [25], Hitachi S-3000N scanning electron microscope (SEM) (Hitachi, Tokyo, Japan) were used to observe the morphology of starch granules. SEM images were distinguished through cross-sections of mature rice seeds including ventral, central, and dorsal area of mature endosperm.

### 4.9. RNA-Seq and Data Analysis

Seeds for both overexpression transgenic rice lines and Zhonghua 11 at 20 DAP were collected for RNA-seq, each group were repeated twice as two biological replicates. Subsequently, RNA was isolated and then high throughput sequencing was performed on BGISEQ-500 platform in Beijing Genomics Institute (BGI; Shenzhen, China). After trimming of low-quality and adaptor sequences from raw sequencing reads, clean data were aligned to *Oryza sativa* ssp. *japonica* cv [26]. Nipponbare genome (IRGSP-1.0, http://rapdb.dna.affrc.go.jp/) [27] using TopHat2 software [28]. The resulted BAM alignment files were subject to Cufflinks to calculate gene expression levels [29]. Differentially expressed genes (DEGs) were determined by Cuffdiff with default parameters, based on the following criteria: (1) the minimum fold-change of gene expression was 2.0; (2) the maximum adjusted *p* value was 0.05 [30]. The RNA-seq data were validated using quantitative real-time PCR analysis for ten randomly selected DEGs. The R package ClusterProfiler were explored to conduct both Gene Ontology (GO) and Kyoto Encyclopedia of Genes and Genomes (KEGG) analysis [31]. The raw sequencing dataset has been submitted to NCBI’s Gene Expression Omnibus (GEO; http://www.ncbi.nlm.nih.gov/geo/) under accession number SRP063765.

### 4.10. Yeast One-Hybrid Assay

Yeast one-hybrid assays were implemented originally according to the Matchmaker^®^ Gold Yeast One-Hybrid Library Screening System User Manual (Clontech). To test the ability of ZmES22 to bind to the core motif CATGT of *GIF1* promoter, CATGT and mutant tandem repeats were cloned and inserted into the *BamH*I and *Hind*III site of the p53/AbAi vector. Yeast Y1HGlod was transformed with the vector pGADT7-ZmES22 and CATGT or mutant tandem repeats plasmids. To evaluate interaction between ZmES22 and the core motif CATGT of *GIF1* promoter, the transformants were screened by plating on SD /-Leu/AbA plates.

## 5. Conclusions

Starch is one of the major components of cereal grains, providing sufficient calories for both human diet and animal feed. Therefore, comprehensive understanding molecular basis of starch synthesis process and its regulatory network is of vital importance. In the present study, we identified a gene *ZmES22*, encoding a typical MADS type transcription factor, which were exclusively highly expressed in maize endosperm, indicating its crucial role in endosperm development of maize. When *ZmES22* was overexpressed in rice, the 1000-grain weight, together with total starch content were remarkably reduced, whereas, the soluble sugar content was significantly higher when compared to wild type. Moreover, overexpression *ZmES22* altered the relative fraction of long branched starch and changed the morphology of starch granule from densely packed, polyhedral starch granules into loosely spherical granules with larger spaces. These results demonstrate that *ZmES22* is a negative regulator that could affect the starch biosynthesis process. Moreover, RNA-seq and qRT-PCR results further illustrated that overexpression of *ZmES22* could downregulate mRNA expression level of numerous key genes in starch synthesis pathway, particularly in early developmental stages in transgenic rice lines. Furthermore, ZmES22 could bind to the promoter region of the *OsGIF1* and downregulate its mRNA expression throughout the endosperm developmental stages. Therefore, we proposed that *ZmES22* might affect starch biosynthesis as well as reducing the rate of grain filling by downregulation of *OsGIF1* in rice. Whether knock-down or knockout of the *ZmES22* gene could contribute to increase of yield in maize remains to be demonstrated.

## Figures and Tables

**Figure 1 ijms-20-00483-f001:**
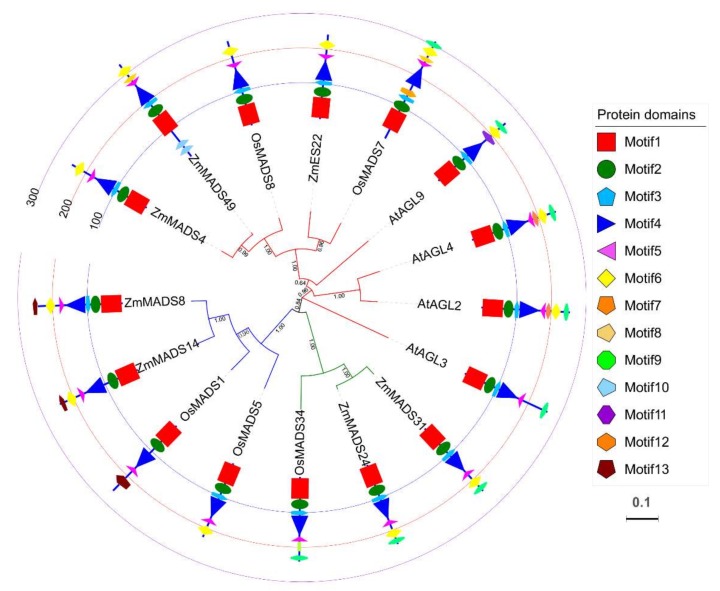
Phylogenetic tree of *ZmES22* homologous genes from maize, rice and Arabidopsis. Phylogenic tree of homologous genes of *ZmES22* from maize, rice and Arabidopsis MADS proteins, which was constructed using conserved MADS domain with MEGA7 software via Neighbor-joining method. Bootstrap value was indicated at each branch point. Gene IDs and predicted functions are listed in Appendix A.

**Figure 2 ijms-20-00483-f002:**
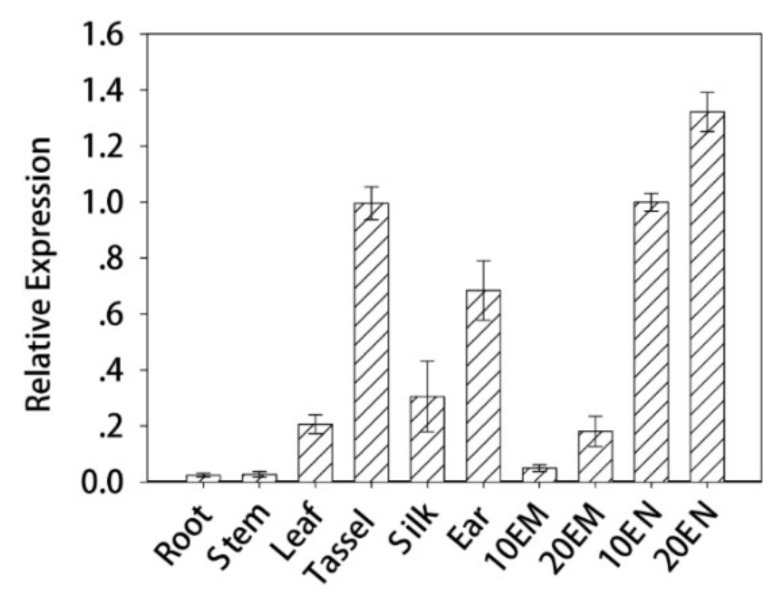
Expression pattern of ZmES22 across diverse tissues. Expression patterns of *ZmES22* in root, stem, leave, tassel, silk, ear, embryo and endosperm was quantified via qRT-PCR. The developmental stage of the embryo and endosperm is indicated by 10 and 20 DAP. Maize *Actin1* was used as the internal control. Error bars are standard deviations of three technical repeats and two biological repeats.

**Figure 3 ijms-20-00483-f003:**
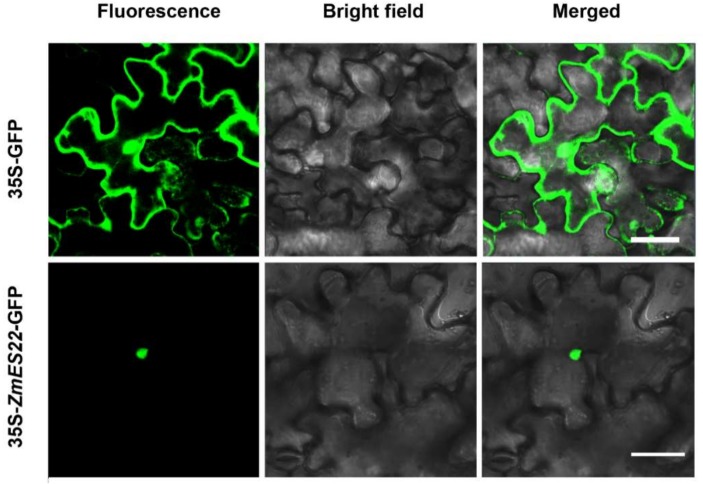
Subcellular localization of ZmES22 in tobacco. The 35S::*ZmES22*-GFP fusion construct and 35S::GFP vector were transiently expressed in tobacco epidermal cells and examined by a confocal laser scanning biological microscope, respectively. Bars = 50 μm.

**Figure 4 ijms-20-00483-f004:**
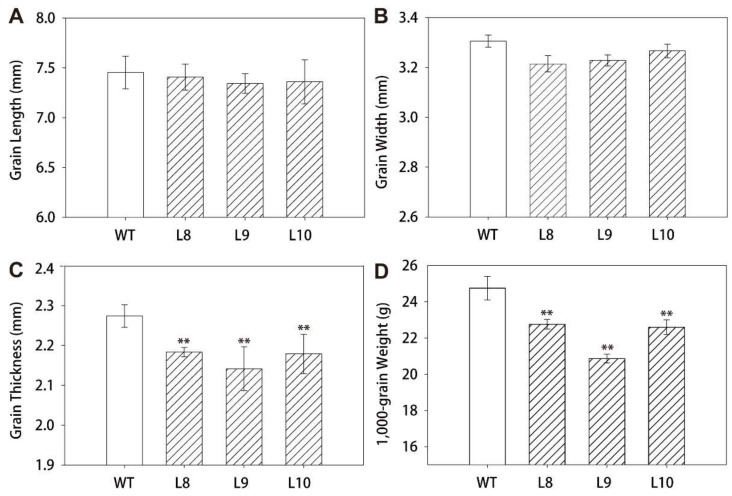
Agronomic characters of seeds from transgenic rice lines that overexpressed *ZmES22*. Grain agronomic characters including grain length (**A**), grain width (**B**), grain thickness (**C**), and 1000-grain weight (**D**) were minutely measured. Data are presented as mean ± SD of three replicates. L: transgenic lines of *ZmES22* seeds; WT: wild-type plants (Zhonghua 11), Student’s *t*-test, ** *p*-value < 0.01.

**Figure 5 ijms-20-00483-f005:**
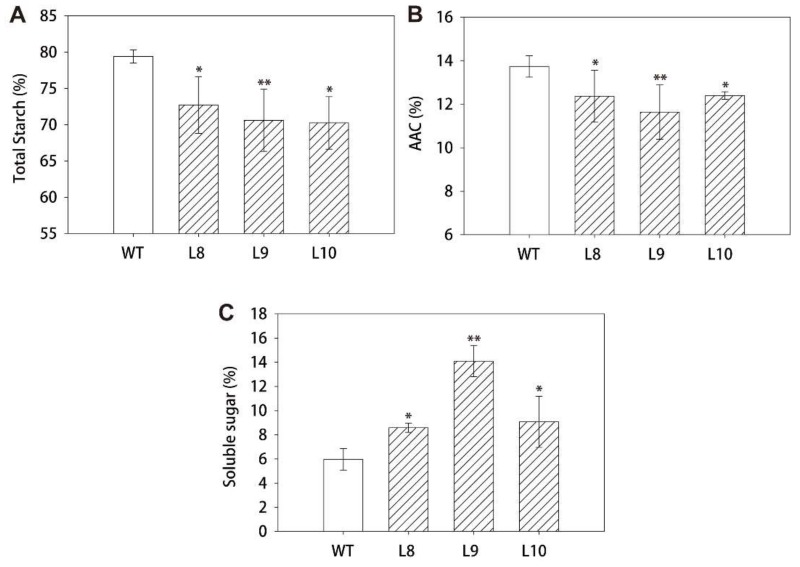
Overexpression of *ZmES22* in rice altered the starch composition. (**A**) Total starch content in rice endosperm. (**B**) Apparent amylose content (AAC) in rice endosperm. (**C**) Soluble sugar content in rice endosperm. Data are presented as mean ± SD of three replicates. L: transgenic lines of *ZmES22* seeds; WT: wild-type plants (Zhonghua 11), Student’s *t*-test, * *p*-value < 0.05, ** *p*-value < 0.01.

**Figure 6 ijms-20-00483-f006:**
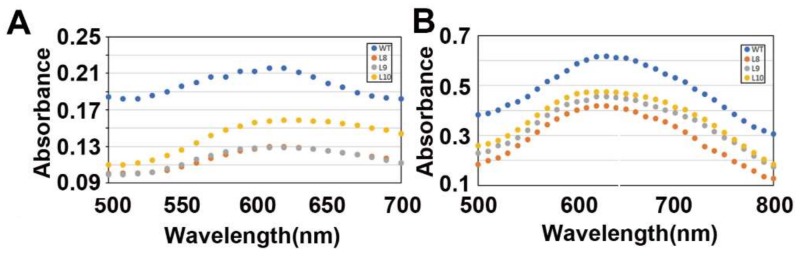
Blue value (BV), maximum absorbance (kmax) and scanning electron microscopy (SEM) images of the transverse sections of transgenic rice seeds. (**A**) BV at 600 nm and kmax represent the ability to combine with iodine. (**B**) BV at 680 nm and kmax represent the ability to combine with iodine. (**C**) Cross-sections of mature seeds are shown in (1). SEM of the ventral area of mature endosperm is shown in a of (2) and indicated by a red square in (1). Bars: 1 mm in (**1**); 10 μm in (2) a: dorsal; b: center; c: belly.

**Figure 7 ijms-20-00483-f007:**
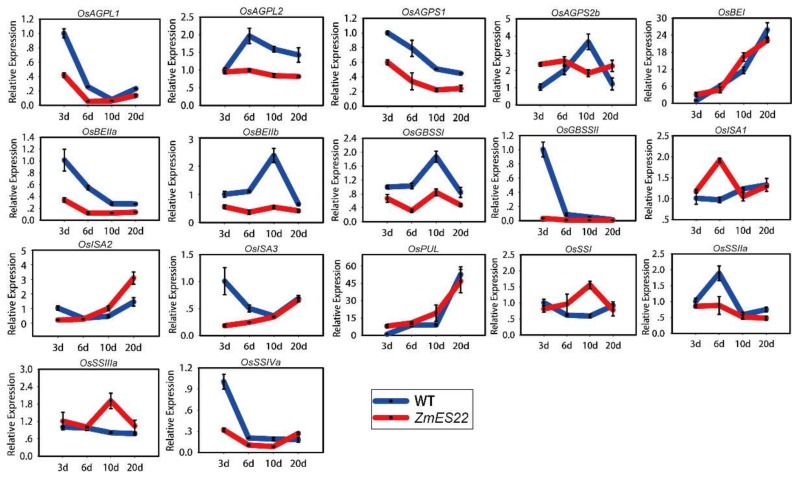
Expression profiles of 17 starch synthesis related genes across diverse developmental stages. Blue line represents wild type (Zhonghua 11), and red line denotes transgenic lines. d is short for days after pollination (DAP). The mRNA expression level of each gene in the three DAP seeds of wild type was used as a control. All data are shown as means ±SD from three biological replicates and two technical replicates. Primers are listed in Appendix A.

**Figure 8 ijms-20-00483-f008:**
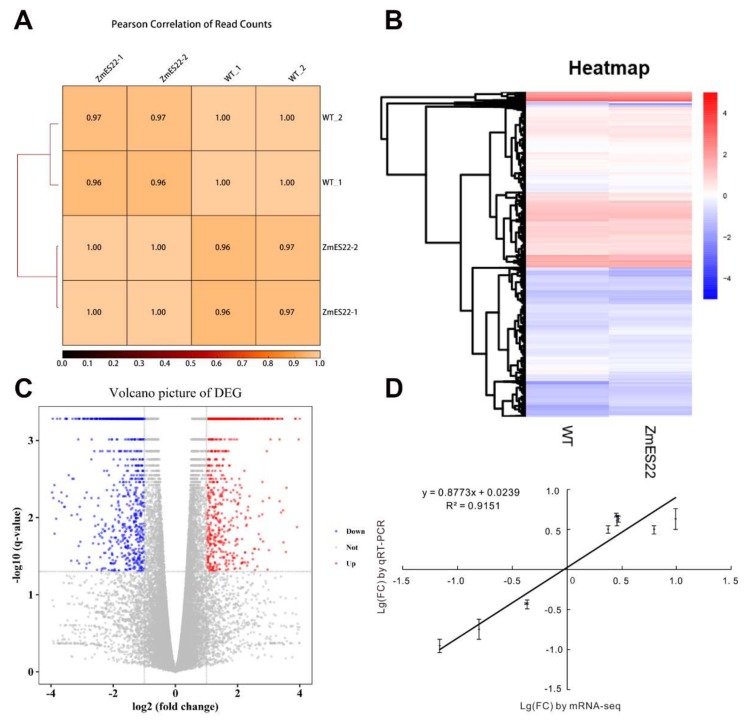
RNA-seq analysis of endosperm at 20 DAP for transgenic rice lines and wild-type plants. (**A**) Pearson correlation of read counts. (**B**) Heat map comparison between Zhonghua 11 and L9. (**C**) A volcano plot of differentially expressed genes (DEGs) about Zhonghua 11 and L9. (**D**) Validation of transcription group data. Expression level changes (log2 (fold change)) of 10 randomly selected DEGs analyzed by RNA-Seq (*x*-axis) were compared with expression data obtained by qRT-PCR (*y*-axis).

**Figure 9 ijms-20-00483-f009:**
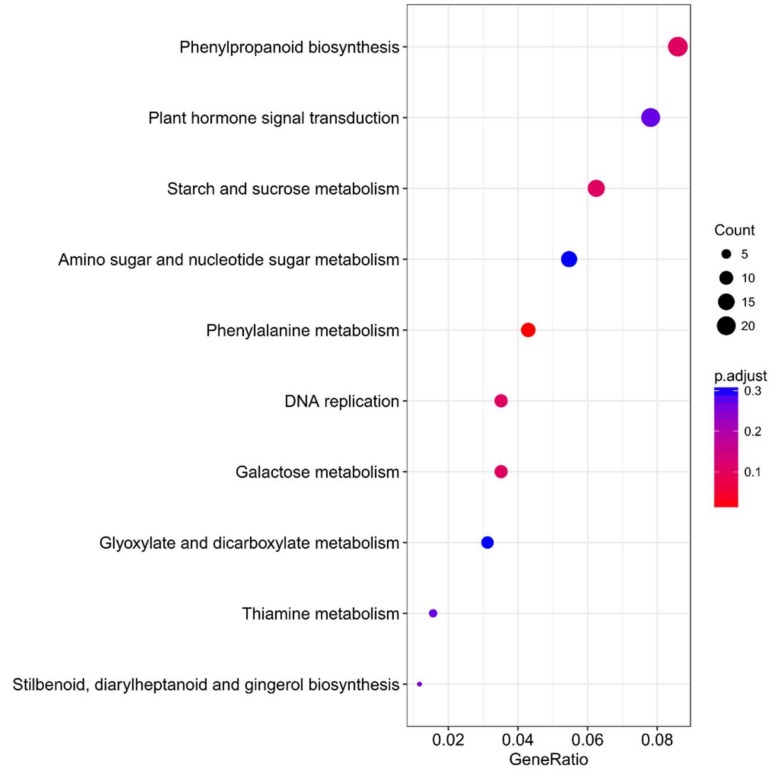
KEGG enrichment analysis for DEGs between transgenic rice lines and wild type plants. KEGG pathways that were enriched for DEGs between transgenic rice lines in comparison to wild type Zhonghua 11. The black circle denotes DEGs that were annotated to one KEGG pathway, and the color panel denoted the *p*-value for each KEGG pathway.

**Figure 10 ijms-20-00483-f010:**
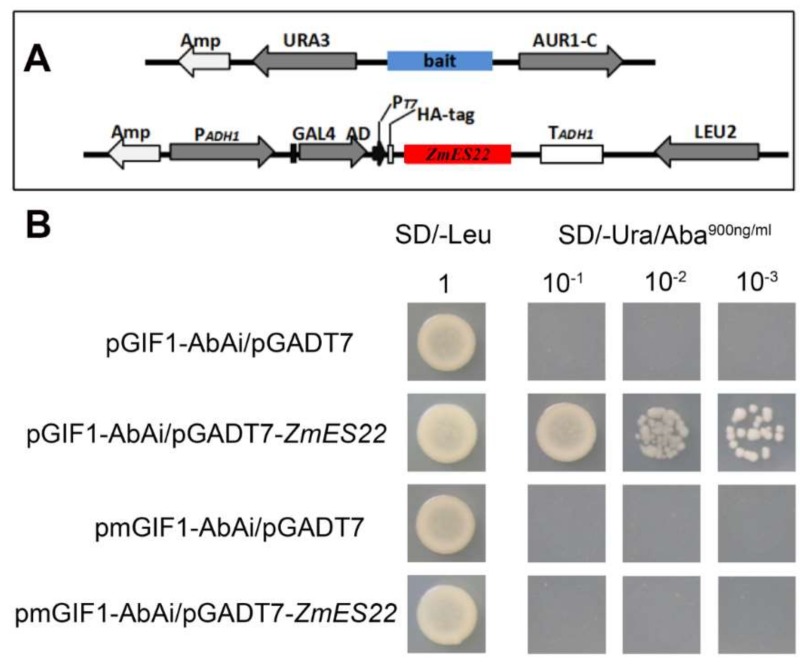
ZmES22 could bind to the core motif of *OsGIF1* via yeast one hybrid assay. (**A**) Schematic structure of yeast expression construct pGAD-*ZmES22* and reporter construct. (**B**) Yeast Y1HGlod was transformed with the vector pGADT7-*ZmES22* and CATGT (*pGIF1*) or mutant tandem repeats (*pmGIF1*) plasmids. The transformants were screened by plating on SD/-Leu/AbA plates to verified the interaction between ZmES22 and the core motif of *GIF1* promoter.

**Table 1 ijms-20-00483-t001:** Statistics of sequencing data

Sample	Clean Reads	Mapped Reads	Clean Base (Gb)	Mapped Base (Gb)	Mapping Rate (%)	Concordant Pair Rate (%)	Q30 (%)	GC Content (%)
WT-1	66,417,130	61,285,695	6.64	6.13	92.27	85.4	94.03	56.95
WT-2	65,482,702	60,254,372	6.55	6.03	92.02	84.7	94.36	57.31
ZmES22-1	65,615,264	60,672,105	6.56	6.07	92.47	85.7	94.15	57.22
ZmES22-2	65,985,744	60,618,187	6.6	6.06	91.9	84.4	92.45	57.01

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
