# Peer review of "Expression of Maize MADS Transcription Factor ZmES22 Negatively Modulates Starch Accumulation in Rice Endosperm"

_ijms, 2019, doi:10.3390/ijms20030483_

Round 1

Reviewer 1 Report

This manuscript describes an investigation of the role of a MADS transcription factor, ZmES22, in starch development.  The most important of the experiments described in the manuscript involve overexpressing the gene in rice.  The authors present evidence that the transcription factor negatively regulates grain filling, primarily by binding to the promoter of the rice GIF1 gene.  The manuscript presents a logical series of experiments that appear to have been carefully conducted.

As someone unfamiliar with most of the details of starch biosynthesis, I found the Introduction to be generally helpful, and the justification for the study was satisfactory.  As a minor criticism, the Introduction focuses primarily on maize, whereas the Discussion focuses mostly on rice.  I could not tell whether the authors were more interested in starch development in maize, rice, or cereals in general.  Perhaps a few sentences about synteny and about the utility of rice as a model for starch biosynthesis in maize would be helpful.

The methods used to characterize agronomic characteristics of the transgenic rice grains were not sufficiently detailed at some points.  For example, how was grain thickness measured – i.e. at the thickest point of the grain?

In the Results, the “analysis of genome-wide transcriptome data” from previous studies is not adequately described in lines 104-105 or in the Materials and Methods. The Figure legends for Figures 1-4 are not properly incorporated into the legends-they are found in the text (lines 127-130, lines 147-150, etc.).

The sentence found in lines 300-302 is ambiguous, as is the description of the same results in lines 222-227.  I felt in general the downregulation of genes involved in starch synthesis was not given as prominent a place in the manuscript as it should have – perhaps a figure showing a summary of these results should be placed in the main body of the manuscript rather than just in the supplemental material.

The English in the manuscript is generally understandable.  There are numerous errors in subject-verb agreement, some use of non-English words (e.g., “detailedly”), sentences in past tense when they should be in present tense, etc.  Additional proofreading is needed.

Author Response

Point 1This manuscript describes an investigation of the role of a MADS transcription factor, ZmES22, in starch development. The most important of the experiments described in the manuscript involve overexpressing the gene in rice. The authors present evidence that the transcription factor negatively regulates grain filling, primarily by binding to the promoter of the rice GIF1 gene. The manuscript presents a logical series of experiments that appear to have been carefully conducted.

As someone unfamiliar with most of the details of starch biosynthesis, I found the Introduction to be generally helpful, and the justification for the study was satisfactory.  As a minor criticism, the Introduction focuses primarily on maize, whereas the Discussion focuses mostly on rice.  I could not tell whether the authors were more interested in starch development in maize, rice, or cereals in general.  Perhaps a few sentences about synteny and about the utility of rice as a model for starch biosynthesis in maize would be helpful.

Response 1: Many thanks for your insightful suggestions. We were intended to clarify the biological function of gene ZmES22, and simultaneously illustrate the molecular mechanism of how this gene participate the starch synthesis process in maize. However, because of difficulty in transgene into maize, we could not get successful transformants in maize. Moreover, previous results revealed that core genes encoding diverse enzymes during the process of starch synthesis exhibited evolutionary conservative ranging from green algae to extant higher plants [1,2]. This similarity indicates that it is possible to clarify the biological function of maize genes in other model plant, such as rice. Furthermore, up to date, enzymes involved in starch synthesis has been soundly documented in rice. Therefore, rice endosperm is a particularly ideal model to screen and identify key transcription factors that could finely tune the process of starch synthesis in maize. In the revised manuscript, we added several sentences and references regarding to the evolutionary history about the core genes in starch synthesis process.

The endosperm is the tissue that most flowering plants produce in the seeds after fertilization. And endosperm development involves the process of starch synthesis and storage protein accumulation. Recent studies revealed that process of starch synthesis was remarkably conserved ranging from green algae to extant higher plants, suggesting that genes encoding starch biosynthesis related enzymes were functionally conserved across diverse lineages [19,20]. Up to date, enzymes involved in starch synthesis has been soundly documented in rice. Therefore, rice endosperm is a particularly ideal model to screen and identify key transcription factors that could finely tune the process of starch synthesis in maize [21].

Point 2The methods used to characterize agronomic characteristics of the transgenic rice grains were not sufficiently detailed at some points.  For example, how was grain thickness measured – i.e. at the thickest point of the grain?

Response 2: Many thanks for your helpful suggestions. We have corrected the methods part as to how to characterize agronomic characteristics of the transgenic rice grains.

Point 3In the Results, the “analysis of genome-wide transcriptome data” from previous studies is not adequately described in lines 104-105 or in the Materials and Methods. The Figure legends for Figures 1-4 are not properly incorporated into the legends-they are found in the text (lines 127-130, lines 147-150, etc.).

Response 3: Many thanks for your comments. We added several sentences to explain the analysis of genome-wide transcriptome data in introduction part.

“In previous studies, a total of 2298 transcription factors were identified and further examined using RNA-seq dataset from 18 representative tissues from maize [11], which provided profound clues regarding to the relationship between development and dynamic expression profiles of key transcription factors. With an emphasize on endosperm-specificity, we identified 36 transcription factors that were preferentially highly expressed in maize endosperm [12].”

We are sorry for the mistakes for the Figure legends for Figure 1-4, and we corrected this mistakes in the revised manuscripts.

Point 4The sentence found in lines 300-302 is ambiguous, as is the description of the same results in lines 222-227.  I felt in general the downregulation of genes involved in starch synthesis was not given as prominent a place in the manuscript as it should have – perhaps a figure showing a summary of these results should be placed in the main body of the manuscript rather than just in the supplemental material.

Response 4: Many thanks for your wonderful suggestion. We placed the supplemental Fig. S6 into the main body of revised manuscript as Fig 7, which summarized the expression profiles for all of the 17 genes in the biosynthesis process of starch during time series after pollination (3 d, 6 d, 10 d and 20 d). Overall, great majority of characterized starch synthesis related genes were down-regulated when compared to the wild type in early endosperm development stages, namely, 3d, 6d and 10d after pollination. However, when we collected samples at 20 DAP, the expression divergences between transgenic lines and wild type could not be significantly detected as early endosperm development stages. Therefore, we deduced that overexpression of ZmES22 might affect mRNA expression of the majority of starch synthesis related genes during the early endosperm developmental stages.

Point 5The English in the manuscript is generally understandable.  There are numerous errors in subject-verb agreement, some use of non-English words (e.g., “detailedly”), sentences in past tense when they should be in present tense, etc.  Additional proofreading is needed.

Response 5: Many thanks for your comments. We are sorry for these mistakes. We changed these non-English words, and used proper tense in the revised manuscript.

References:

1.   Deschamps, P.; Colleoni, C.; Nakamura, Y.; Suzuki, E.; Putaux, J.-L.; Buleon, A.; Haebel, S.; Ritte, G.; Steup, M.; Falcon, L.I.; et al. Metabolic Symbiosis and the Birth of the Plant Kingdom. Molecular Biology and Evolution 2008, 25, 536–548.

2.   Qu, J.; Xu, S.; Zhang, Z.; Chen, G.; Zhong, Y.; Liu, L.; Zhang, R.; Xue, J.; Guo, D. Evolutionary, structural and expression analysis of core genes involved in starch synthesis. Scientific Reports 2018, 8.

Reviewer 2 Report

1- The introduction need to be improved and organized more to be more clear.

2- the part of RNA sequence in the introduction needs more explanation and references

3- Results can be improved 

4- The Results rely on a lot of supplemental materials and figures that It was not provided with the manuscript

5- The statistical analysis for the data was very confusing especially with using different levels of p value

6-Figure 6 Legend and subtitle was too long

I recommend you either to divide the figure into two or shorten the Subtitle 

there is a lot of unnecessary details

The EM bar can be shown on the picture no need to rewrite it in the subtitle

7-Conclusion

the manuscript lack a good conclusion part

just short summary that doesn't relate much on the results

Line 327 starts with uncomplete sentence; In summary...….

Author Response

Point 1The introduction needs to be improved and organized more to be clearer.

Response 1: Thanks for your helpful suggestions. In the revised manuscript, the brief introduction of MADS type transcription factors were deleted from Introduction Part and these sentences were simplified as opening words in part 1 of Results. Moreover, we added detailed explanation of the RNA sequencing and the reason why we choose ZmES22 as the target genes. The newly organized manuscript was proposed to be logically and easier to be understood than previous one.

Point 2the part of RNA sequence in the introduction needs more explanation and references

Response 2: Many thanks for your suggestions. The RNA sequencing dataset were initially performed to identify key transcription factors that were expressed in 18 representative tissues, for example, root, shoot, leaf, ear, tassel and kernel, which could provide important clues for understanding the key regulators in diverse developmental stages of maize [1]. The research of our lab focused mainly on what transcription factors were responsible for regulation of starch synthesis process in maize. Therefore, we collected 36 genes, which exhibited exclusively high expression level in endosperm, indicating that they might play an important role in regulation of endosperm development in maize [2]. The gene, ZmES22, were one of the 36 endosperm-specific transcription factors. In the revised manuscript, we added these references and clarified the RNA sequencing part.

In previous studies, a total of 2298 transcription factors were identified and further examined using RNA-seq dataset from 18 representative tissues from maize [11], which provided profound clues regarding to the relationship between development and dynamic expression profiles of key transcription factors. With an emphasize on endosperm-specificity, we identified 36 transcription factors that were preferentially highly expressed in maize endosperm [12].

Point 3 Results can be improved 

Response 3: Many thanks for your suggestions. In the revised manuscript, we placed the supplemental Fig. S6 into the main body of revised manuscript as Fig 7, which summarized the expression profiles for all of the 17 characterized genes in the biosynthesis process of starch during time series after pollination (3d, 6d, 10d and 20d). This figure could help to illustrate the regulatory effect for ZmES22 gene when overexpressed in rice.

Point 4The Results rely on a lot of supplemental materials and figures that was not provided with the manuscript.

Response 4: Many thanks for your comments. The results part indeed rely on a lot of supplemental materials and figures, which are appended along with the revised manuscript.

Point 5The statistical analysis for the data was very confusing especially with using different levels of p value.

Response 5: Many thanks for your comments. We carefully checked the statistical analysis explored in the present study, and simultaneously we corrected the wrong usage of p value in revised manuscript.

Point 6Figure 6 Legend and subtitle was too long, I recommend you either to divide the figure into two or shorten the Subtitle. There is a lot of unnecessary details and the EM bar can be shown on the picture no need to rewrite it in the subtitle.

Response 6: Many thanks for your helpful suggestions. We deleted the unnecessary details for the legends of figure 6. It is now brief and to the point in revised manuscript.

Point 7Conclusion

the manuscript lacks a good conclusion part, just short summary that doesn't relate much on the results

Response 7: Many thanks for your suggestions. We rewrote the conclusion part and added it behind the Materials and Methods part in the revised manuscript.

“starch is one of the major components of cereal grains, providing sufficient calories for both human diet and animal feed. Therefore, comprehensive understanding molecular basis of starch synthesis process and its regulatory network is of vital importance. In the present study, we identified a gene ZmES22, encoding a typical MADS type transcription factor, which were exclusively highly expressed in maize endosperm, indicating its crucial role in endosperm development of maize. When ZmES22 was overexpressed in rice, the 1000-grain weight, together with total starch content were remarkably reduced, whereas, the soluble sugar content was significantly higher when compared to wild type. Moreover, overexpression of ZmES22 altered the relative fraction of long branched starch and changed the morphology of starch granule from densely packed, polyhedral starch granules into loosely, spherical granules with larger spaces. These results demonstrate that ZmES22 is a negative regulator that could affect the starch biosynthesis process. Moreover, RNA-seq and qRT-PCR results further illustrated that overexpression of ZmES22 could downregulate mRNA expression level of numerous key genes in starch synthesis pathway, particularly in early developmental stages in transgenic rice lines. Furthermore, ZmES22 could bind to the promoter region of the OsGIF1 and downregulate its mRNA expression throughout the endosperm developmental stages. Therefore, we proposed that ZmES22 might affect starch biosynthesis as well as reduce the rate of grain filling by downregulating OsGIF1 in rice. Whether knock-down or knockout of the ZmES22 gene could contribute to increase of yield in maize remains to be demonstrated.”

Point 8Line 327 starts with uncomplete sentence; In summary...….

Response 8: Many thanks for your comments. We were sorry for this mistake, and we rewrote this sentence and additional proofreading was performed throughout the revised manuscript.

References:

1.   Jiang, Y.; Zeng, B.; Zhao, H.; Zhang, M.; Xie, S.; Lai, J. Genome-wide Transcription Factor Gene Prediction and their Expressional Tissue-Specificities in Maize. Journal of Integrative Plant Biology 2012, 54, 616–630.

2.   Cai, H.; Chen, Y.; Zhang, M.; Cai, R.; Cheng, B.; Ma, Q.; Zhao, Y. A novel GRAS transcription factor, ZmGRAS20, regulates starch biosynthesis in rice endosperm. Physiology and Molecular Biology of Plants 2017, 23, 143–154.